# Reactive Transport and Removal of Nutrients and Pesticides in Engineered Porous Media

**Dongli Tong [1,2], Jie Zhuang [1,3] and Xijuan Chen [1,*]**

[1]   Key Laboratory of Pollution Ecology and Environmental Engineering, Institute of Applied Ecology, Chinese Academy of Sciences, Shenyang 110016, Liaoning, China

[2]   University of Chinese Academy of Sciences, Beijing 100039, China

[3]   Department of Biosystems Engineering and Soil Science, Center for Environmental Biotechnology, University of Tennessee, Knoxville, TN 37996, USA

*   Correspondence: chenxj@iae.ac.cn

**Abstract:** Agricultural nonpoint pollution has been recognized as a primary source of nutrients and pesticides that contaminate surface water and groundwater. Reactive materials have great potential to remove nutrients and pesticides from agricultural drainage water. In this study, we investigated the reactive transport and removal of coexisting nitrate, phosphate, and three pesticides (tricyclazole, isoprothiolane, and malathion) by iron filings and natural ore limestone through column experiments under saturated flow conditions. Breakthrough results showed that 45.0% and 35.8% of nitrate were removed by iron filings and limestone during transport, with average removal capacities of 2670 mg/kg and 1400 mg/kg, respectively. The removal of nitrate was mainly due to microbial denitrification especially after 131–154 pore volumes (≈30 d), whereas reduction to ammonia dominated nitrate removal in iron filings during early phase (i.e., <21.7 d). The results showed that 68.2% and 17.6% of phosphate were removed by iron filings and limestone, with average removal capacities of 416.1 mg/kg and 155.2 mg/kg, respectively. Mineral surface analyses using X-ray diffraction (XRD) and scanning electron microscope (SEM) coupled with energy-dispersive X-ray analysis (EDX) suggested that ligand exchange, chemical precipitation, and electrostatic attraction were responsible for phosphate removal. Chemical sorption was the main mechanism that caused removals of 91.6–100% of malathion and ≈27% of isoprothiolane in iron filings and limestone. However, only 22.0% and 1.1% of tricyclazole were removed by iron filings and limestone, respectively, suggesting that the removal might be relevant to the nonpolarity of tricyclazole. This study demonstrates the great potential of industrial wastes for concurrent removal of nutrients and pesticides under flow conditions.

**Keywords:** nutrient; pesticide; removal; reactive transport; nonpoint pollution

## 1. Introduction

Nitrogen and phosphorus contamination in water has become an environmental and public health problem worldwide and frequently results in eutrophication with potential health hazards to both animal and humans [1,2]. Agricultural drainage/runoff is recognized as a primary source of increased nitrogen and phosphorus in aquatic systems. Bouraoui and Grizzetti [1] reported that agriculture attributed to ≈55% of nonpoint water pollution of eutrophic surface water in Europe. Excessive application and low utilization of agricultural fertilizers increase the transport of nitrogen and phosphorus from farmlands to surface water and groundwater. For instance, nitrate, and phosphate concentration in water discharged from agricultural filed were reported to be 1.1–23.8 mg-N/L and 0.01–0.26 mg-P/L in Northwest Indiana, USA [3]. A similar study indicated that soluble phosphorous concentration in water from agricultural fields reached 1.0 mg-P/L [4]. Additionally,

pesticides are extensively used in agriculture to boost crop production, which can be carried with runoff to surface water or leached downward into groundwater, resulting in deterioration of water quality and increasing the potential risk to human and environmental health (e.g., toxic, carcinogenic, and mutagenic effects) [5]. Papadakis et al. [6] reported that the detection frequency of pesticides was 42% in rives and agriculture drainage canals around Lake Vistonis Basin in Greece, and the risk assessment revealed that significant ecological risk towards the aquatic organisms in more than 20% of water samples.

The most commonly used technologies in agriculture runoff remediation including ecological ditch [7], ecological floating bed [8,9], and constructed wetland [10,11]. These are all ecological friendly technologies with high biodiversity. Removal of nutrients and pesticides is achieved through vegetation uptake, microbial degradation or utilization, and flocculation or sedimentation facilitated by biofilms adhering to plant roots [7,9]. However, growth of plants is seasonal, and remediation is dependent on growth rate of plants [11]. Moreover, secondary pollution of nutrient may occur due to decomposition of dead plants [8]. Sorption by reactive materials has been widely used and becomes one of the promising technologies for nutrients and pesticides removal in the last decades due to its advantages such as high availability of adsorption materials, large adsorption capacity, high removal rates, reusability of materials, and ease of operation [12–14].

Sorption process by various reactive materials has been found successful in removing different types of compounds, including heavy metals [15], fluoride [16], nitrate [13,17], phosphate [14,18], chlorinated organic compounds [19], polychlorinated biphenyls [20], and pesticides [12,21]. Selection of appropriate reactive materials for the specific types of pollutants is significant to achieve large removal capacity. Various reactive materials have been investigated for removal of nutrients and/or pesticides [12,22]. For example, batch studies showed that 97–99% of phosphate (10 mg-P/L) was removed in 5–60 min by a nano-zero valent iron [18,23]. Nitrate reduction and norfloxacin oxidation could be successfully achieved in a nano-zero valent iron and ultrasound irradiation process, with the removal efficiencies of nitrate and norfloxacin reaching nearly 96% and 94% within 120 min, respectively [17]. García-Jaramillo et al. [24] reported that biochar from alperujo compost at 400–700 °C under oxygen-limited conditions had high tricyclazole removal affinity (55.9–90.3%). Previous studies were mostly focused on the synthesis or modification of materials for removal of nutrients or pesticides [13,17,23]. However, the synthetic/modified materials are relatively expensive and complicated for in situ application to direct decontamination of agricultural drainage. In addition, those reported results were mostly based on batch experiments, representing static removal processes in closed reactors [13,25]. Therefore, the derived information may not represent the practical efficiency under continuous flow conditions. Compared to batch experiments, flow-through column experiments are closer to the operational conditions, in which agricultural drainage water continuously flows through treatment systems packed with different porous solid materials [26]. Obviously, extensive evaluation is needed to determine the overall feasibility of reactive materials under flow conditions for a long time.

In this study, two low cost reactive materials, industrial iron filings and natural ore limestone, were tested due to their expected high removal capacity, economic feasibility, high permeability, and general availability to agricultural communities. Three pesticides widely used in rice fields, including tricyclazole, isoprothiolane, and malathion, were tested, considering their frequent detection and relative high concentrations in surface water [6,27]. The objective of this study was to investigate the potential and mechanisms of selected materials in removing nitrate, phosphate and pesticides from synthetic agricultural drainage under flow conditions. Long duration column experiments were carried out to determine the viability of the systems. The results are expected to provide significant insights into the construction of cost-effective, environmentally friendly, portable reactive filter systems for reducing nonpoint agricultural drainage pollution.

## 2. Materials and Methods

### 2.1. Materials

A natural ore limestone obtained from Liaoning Xiuyan Qinghua Mining Co. Ltd. (Yingkou, China) was ground to 0.30–1.00 mm prior to use. Iron filings with irregular shape and sizes (0.42–0.85 mm) as a steel by-product were collected from a metal machining factory located in Zhengzhou China and used directly without chemical pretreatment. Figures 1 and 2 show XRD spectra, SEM photo graphs, and energy-dispersive X-ray analysis (EDX) mapping of limestone and iron filings. The composition of natural limestone and iron filings are very close to the formula calcite ($CaCO_3$) and zero valent iron according to XRD spectra, respectively. The specific surface area of materials was determined by a Brunauer–Emmett–Teller (BET) surface area analyzer (ASAP 2460, Micromeritics, Norcross, GA, USA). The limestone and iron filings had specific surface area of 0.25 $m^2$/g and 0.14 $m^2$/g, respectively. Quartz sand (0.18–0.25 mm) was obtained from Tianjin Kermio Chemical Reagent Co. Ltd. (Tianjin, China). Prior to use, the quartz sand was washed with 10 mM HCl for 24 h and 10 mM NaOH for 24 h to remove of impurities, and then rinsed to neutral with deionized water and dried at 105 °C for 6 h.

The synthetic agricultural drainage was prepared by mixing $KH_2PO_4$ (13.40 mg/L), $Na_2HPO_4$ (22.20 mg/L), $KNO_3$ (216.64 mg/L), NaCl (442.55 mg/L), tricyclazole (1.00 mg/L), isoprothiolane (1.00 mg/L), and malathion (1.00 mg/L) [3,28]. The resulting total ionic strength of the solution was 10 ± 0.2 mM. The initial pH value was adjusted to 6.7 ± 0.2 by using 0.1 M HCl or 0.1 M NaOH. The pesticides were purchased from AccuStandard Inc (New Haven, CT, USA) and other chemicals were purchased from Tianjin Kermio Chemical Reagent Co. Ltd. (Tianjin, China) in analytical grade.

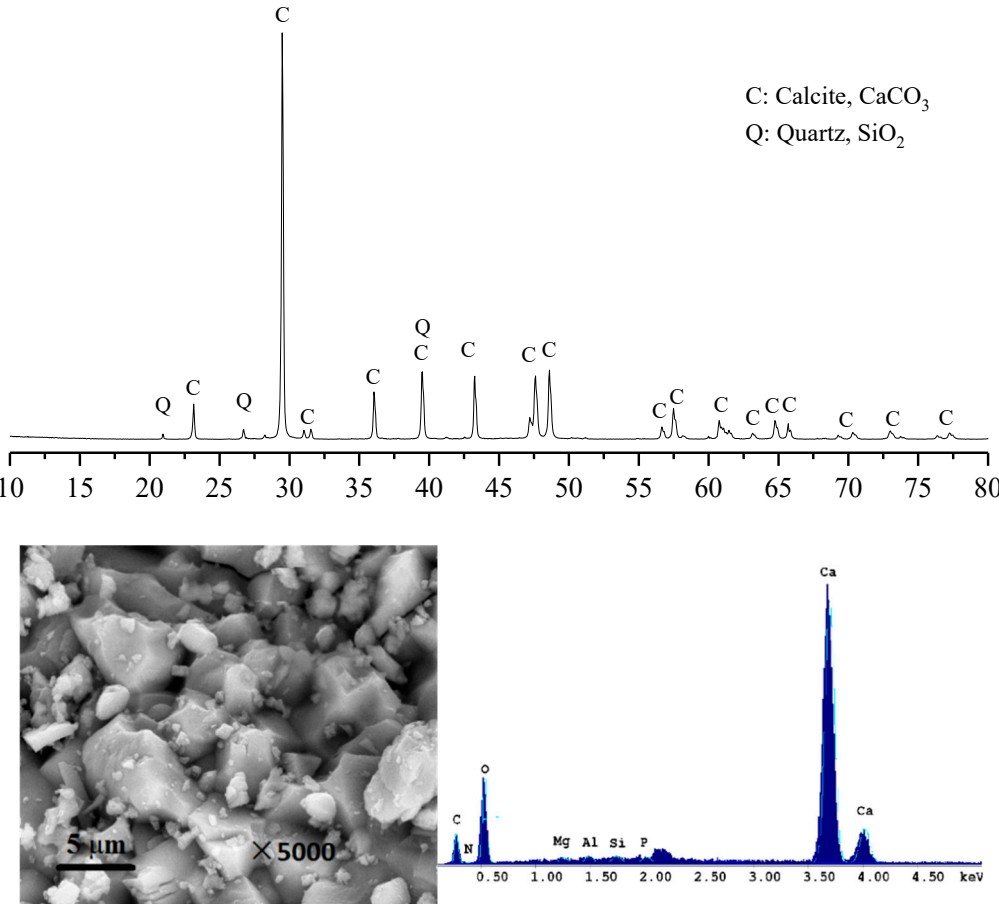

**Figure 1.** XRD spectra, SEM image, and energy-dispersive X-ray mapping (EDX) of limestone.

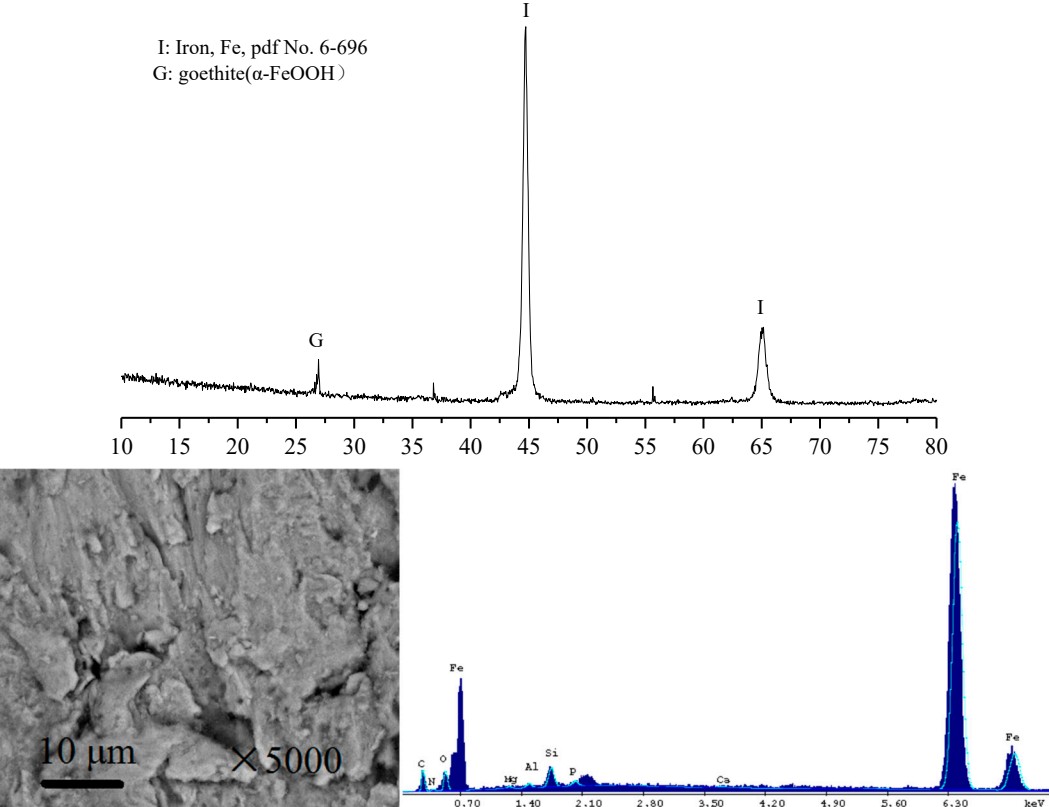

**Figure 2.** XRD spectra, SEM image, and energy-dispersive X-ray mapping (EDX) of iron fillings.

## 2.2. Column Experiment

The experimental system is shown in Figure 3. Stainless steel columns (15.8 cm in depth; 5 cm in inner diameter) were used in the study. Solution inlet was set at the bottom of the column, and the outlet was set at the top of the column. Reactive zone of iron filings or limestone was located in the middle 5 cm of the column. A lower support layer of 5 cm of quartz sand was packed at the bottom of the column and another 5 cm layer of quartz sand was placed on the top to ensure uniform cross-sectional flow of the influent and to prevent reactive material loss from the column. A column totally packed with quartz sand was used as control column. Nylon membranes (30 µm in pore size) were placed at the top and bottom of the column and between the quartz sand and the reactive material to avoid any loss or mixture of the packing materials. The columns were continuously fed with a synthetic agricultural drainage solution at a pore velocity of ≈1.0 cm/h by peristaltic pumps (BT100-2J, DG-4 pump head, LongerPump, Baoding, China). The average hydraulic residence time (HRT) of the influent solution was ≈5 h. The total volume of influent for each column was 39.9 ± 8.0 L (750 pore volumes). The effluent samples were collected on regular time intervals for analysis. Each experiment was performed in replicate to examine the reproducibility of results. The difference in breakthrough curves was statistically analyzed using one-way analysis of variance. Data from one column was plotted to compare the roles of different reactive materials, considering data from two columns differ at $p < 0.05$. The experimental conditions of all columns are summarized in Table 1.

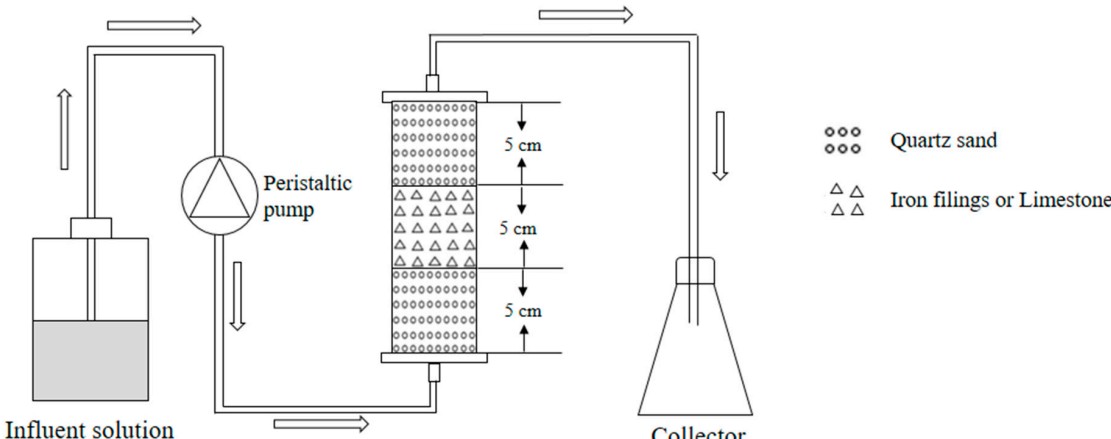

**Figure 3.** Column system of transport experiment.

**Table 1.** The experimental conditions of columns.

| Exp. # | Materials | NaN$_3$ Input | Bulk Density (g/cm$^3$) | Porosity (%) | Pore Velocity (cm/h) | Hydraulic Residence Time (h) |
|---|---|---|---|---|---|---|
| 1 | Quartz sand | No | 1.62 | 39.05 | 1.06 | 5.0 |
| 2 | Iron fillings | No | 3.25 | 58.47 | 1.09 | 4.9 |
| 3 | Iron fillings | No | 3.16 | 59.64 | 0.96 | 5.2 |
| 4 | Iron fillings | Yes | 3.05 | 61.04 | 1.02 | 5.0 |
| 5 | Limestone | No | 1.62 | 39.74 | 1.12 | 4.5 |
| 6 | Limestone | No | 1.57 | 41.46 | 1.02 | 5.0 |
| 7 | Limestone | Yes | 1.60 | 40.64 | 1.10 | 5.0 |

## 2.3. Solid Characterizations

The packed reactive materials were collected immediately after the column experiments at two depths: (5–7.5 cm and 7.5–10 cm from the inlet). The collected solid samples were freeze-dried and stored in closed containers until analysis. X-ray direction (XRD) analysis was carried out to identify crystallographic structures of the samples using a computer-controlled X-ray diffractometer (Rigaku D/max 2400, Rigaku, Tokyo, Japan). The surface morphology was analyzed using a scanning electron microscope (SEM, Quanta 250, FEI, Hillsborough, OR, USA) coupled with energy-dispersive X-ray analyzer (EDX, AMETEK, San Diego, CA, USA). The specific surface area of reactive materials were measured using a specific surface area analyzer (ASAP 2460, Micromeritics, Norcross, GA USA). The measured data were compared with those of the original materials to identify the mechanisms responsible for the removal of nitrate and phosphate.

## 2.4. Analysis of Aqueous Samples

The concentrations of nitrate and phosphate were determined spectrophotometrically by an UV-visible spectrophotometer (UH5300, Hitachi, Tokyo, Japan) according to Chinese Environment Standards HJ/T 346-2007 and HJ/T 346-2007. Aqueous pesticide concentrations were quantified with a high performance liquid chromatography (HPLC) (1260 LC, Agilent, Santa Clara, CA, USA) equipped with an ultraviolet detector and a ZORBAX SB-C18 analytical column (150 × 4.6 mm id, 5 μm) using methanol–water (80:20, v/v) as mobile phase at a flow rate of 0.2 mL/min (35 °C).The ultraviolet wavelength was set at 220 nm, and the injection volume was fixed at 20 μL. External standards were used for quantification of pesticides. The method detection limit was 0.005, 0.02, and 0.05 mL/min for trycyclazole, isoprothiolane, and malathion, respectively. The effluent pH was measured with pH meter (PB-10, Sartorius, Gottingen, Germany).

The integration tools within Origin 2017 (OrginLab Corp., Northampton, MA, USA) was used to calculate the cumulative removal rate (%) and average removal capacity (mg/kg) of nutrients or pesticides during the transport experiments.

## 3. Results and Discussion

### 3.1. Removal of Nitrate

The nitrate breakthrough results from the columns packed with quartz sand, iron filings and limestone are illustrated in Figure 4. The nitrate broke through the quartz sand column after the first pore volume, and then its relative effluent concentration ($C/C_0$) increased quickly to ≈1 at ≈2 pore volumes. The average maximum $C/C_0$ was ≈1 before 24.0 pore volumes (5.0 d), indicating that no nitrate removal (<3%). However, a gradual decrease in nitrate $C/C_0$ and an increase in cumulative removal rate took place. Nitrate $C/C_0$ decreased from ≈1 at 24.0 pore volumes to 0.70 at 175 pore volumes (36.5 d), while cumulative removal rate increased to 19.2%, yielding an average removal capacity of 230.5 mg-N/kg (Table 2). This removal was primarily caused by biological denitrification [25,29].

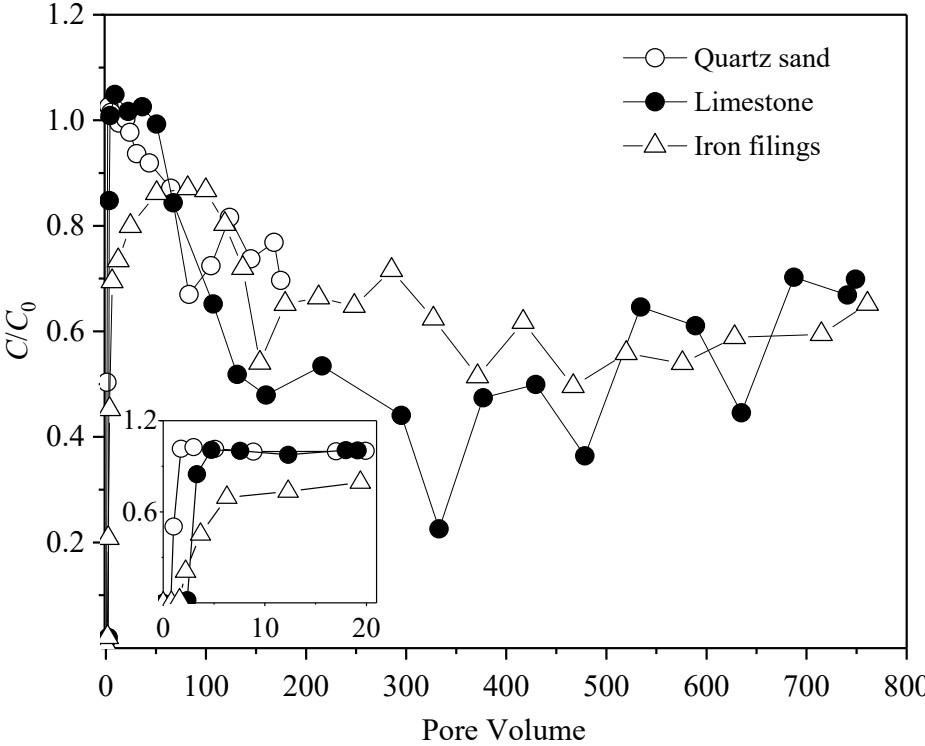

**Figure 4.** Nitrate breakthrough from columns packed with different reactive materials. Subgraph is detailed view of the results within the first 20 pore volumes.

**Table 2.** Cumulative removal of nitrate, phosphate and pesticides by reactive materials within 750 pore volumes (156.3 d) and by quartz sand with 175 pore volumes (36.5 d).

| Column | Nitrate | | Phosphate | | Tricyclazole | | Isoprothiolane | | Malathion | |
|--------|---------|-----|-----------|------|--------------|-------|----------------|-------|-----------|--------|
| | mg/kg | % | mg/kg | % | mg/kg | % | mg/kg | % | mg/kg | % |
| Quartz sand | 230.46 | 19.20 | 1.77 | 0.95 | 0 | 0 | 4.60 | 11.10 | 5.19 | 17.44 |
| Limestone | 2670.35 | 45.03 | 155.21 | 17.58 | 2.22 | 1.11 | 51.81 | 26.97 | 166.77 | 91.63 |
| Iron fillings | 1400.37 | 35.83 | 416.05 | 68.20 | 30.07 | 22.02 | 35.23 | 26.96 | 138.91 | 100.00 |

Similar to quartz sand, nitrate broke through the limestone column at ≈2 pore volumes. Nitrate $C/C_0$ increased quickly to ≈1 at 4.3 pore volumes (0.9 d) and then stabilized between 4.3 pore volumes (0.9 d) and 50.6 pore volumes (10.5 d). The cumulative nitrate removal rate and average removal capacity were 3.7% and 68.6 mg-N/kg within the first 50.6 pore volumes (10.5 d), indicating minor nitrate removal capacity. Afterwards, effluent nitrate $C/C_0$ decreased and reached an average of 0.53 at 131 pore volumes (27.3 d). Meanwhile, the cumulative nitrate removal rate of limestone increased

from 3.7% to 45.0%, higher than that of quartz sand (19.2%). This result suggests that limestone may require relatively long operation times (e.g., >27.3 d) to fully develop the capability to remove nitrate. The nitrate average removal capacity by limestone gradually increased from 68.6 mg-N/kg to 2670 mg-N/kg between 50.6 pore volumes (10.5 d) and 750 pore volumes (156.3 d) (Table 2). Cationic species (e.g., $Ca^{2+}$, $CaHCO_3^+$, and $CaOH^+$) on the limestone have been proved to be able to adsorb nitrate by electrostatic attraction [30]. On the other hand, nitrate may degrade biologically after 50.6 pore volumes (10.5 d) [25,29]. However, the former mechanism may be negligible as indicated by the complete breakthrough of nitrate from the limestone within the first 4.3 pore volumes (0.9 d). A similar result was reported by Cui et al. [31] that zeolite and limestone as substrates in ecological floating beds might enhance the nutrients removal rate by promoting the growth of microorgamisms (46–54%), rather than their absorption effect (only 2.3–3.9%).

In iron filings column, nitrate breakthrough curves had a similar trend to that in limestone (Figure 4). However, complete breakthrough was not observed. Nitrate broke through iron filings column at ≈1 pore volume, and then its $C/C_0$ increased and stabilized at 0.87 between 46.7 pore volumes (9.7 d) and 104.3 pore volumes (21.7 d). The cumulative removal rate was 19.3% and average removal capacity was 108.6 mg-N/kg during the first 104.3 pore volumes (21.7 d). This is consistent with the observation by Westerhoff and James [32], who reported that ≈20% of nitrate was removed by scrap iron filings packed column within ≈200 pore volumes. This removal is attributed to chemical reduction and sorptbion by iron and its corrosive products [33–35]. In $Fe^0$-$H_2O$ system, iron is an electron donor, while nitrate serves as an electron acceptor. The potential reactions between iron and nitrate can be described by the following chemical equations [32,33,35,36].

$$5\,Fe^0 + 2NO_3^- + 6\,H_2O \rightarrow 5\,Fe^{2+} + N_2 + 12\,OH^- \tag{1}$$

$$4\,Fe^0 + NO_3^- + 7\,H_2O \rightarrow 4\,Fe^{2+} + NH_4^+ + 10\,OH^- \tag{2}$$

$$Fe^0 + NO_3^- + H_2O \rightarrow Fe^{2+} + NO_2^- + 2\,OH^- \tag{3}$$

$$2\,Fe^{2+} + NO_3^- + 13\,H_2O \rightarrow 8\,FeOOH + NH_4^+ + 14\,H^+ \tag{4}$$

$$12\,Fe(OH)^+ + NO_3^- + H_2O \rightarrow 4\,Fe_3O_4 + NH_4^+ + 10\,H^+ \tag{5}$$

$$4\,Fe^0 + NO_3^- + 10\,H^+ \rightarrow 4\,Fe^{2+} + NH_4^+ + 3\,H_2O \tag{6}$$

$$Fe^0 + NO_3^- + 2\,H^+ \rightarrow Fe^{2+} + NO_2^- + H_2O \tag{7}$$

As the flow experiments proceeded, iron corrosive products (such as lepidocrocite, goethite, magnetite, and green rust I/II) might adsorb nitrate from the flowing solution via electrostatic attraction [35,37]. Additionally, significant iron corrosion was indicated by the hundredfold increase in specific surface area of iron filings from 0.25 $m^2$/g (before the experiments) to 18–36 $m^2$/g (after the experiments), suggesting that more sorption sites became available for nitrate removal during the transport process. A few studies indicated that nitrite, nitrogen gas, and ammonia could be possible products of nitrate reduction [32,36]. To confirm the ammonium production, we monitored the concentrations of nitrate and ammonium in the influent and effluent from iron filings columns during 104.3 pore volumes (21.7 d). The produced ammonium is plotted against the reduced nitrate in Figure 5. The slope value of the correlation indicates that 73.9% of the added nitrate was reduced into ammonium. This result is consistent with the laboratory column tests by Westerhoff and James [32], who reported 70% of the added nitrate was recovered as ammonium. After 104.3 pore volumes (21.7 d), nitrate $C/C_0$ decreased and reached an average of 0.60 at 153.8 pore volumes (32.0 d), suggesting that iron filings required relatively long operation times (e.g., >32.0 d) to fully develop the capability of nitrate removal. The cumulative nitrate removal rate of iron filings increased to 35.8%, while the average removal capacity increased to 1400 mg-N/kg within 750 pore volumes, which was 1.9 times lower than that of limestone (2670 mg/kg) (Table 2). This result could be attributed to microbial denitrification [25,29] since no antibacterial measures were applied during the long-time column experiments.

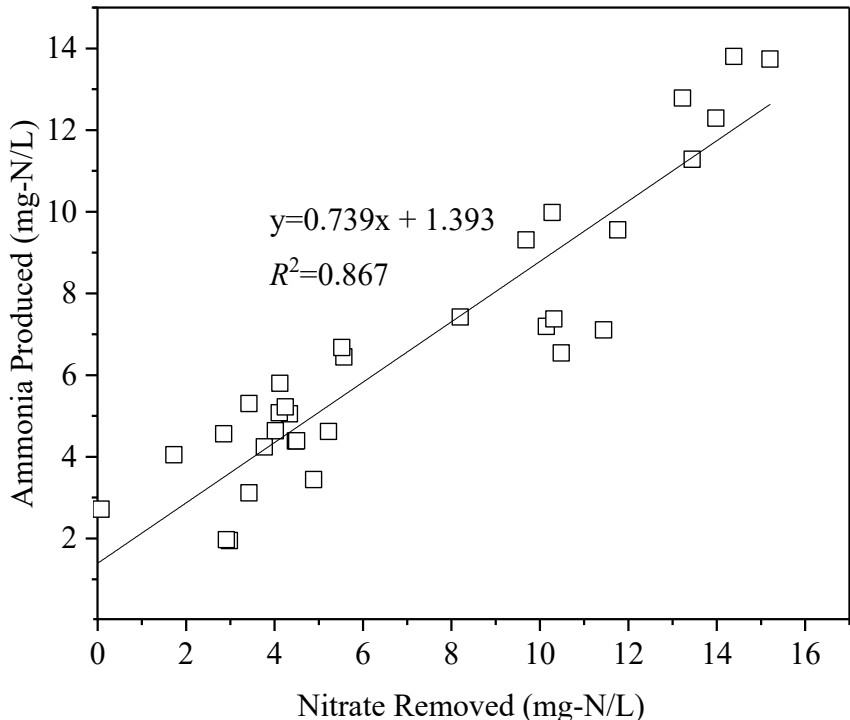

**Figure 5.** Correlation between nitrate-N removal and ammonia-N production in the effluent of iron fillings columns within 104.3 pore volumes (21.7 d).

To verify microbial denitrification during transport, an additional set of column experiemnts packed with iron fillings and limestone were carried out with $NaN_3$ (200 mg/L) added to the influent for inhibiting microorganism growth. Figure 6 demonstrates that there was significant difference in the effluent concentration of nitrate between the columns with and without $NaN_3$ after 50.6 pore volumes (10.5 d) in limestone, and 2.8 pore volumes (0.4 d) in iron fillings. Nitrate $C/C_0$ in limestone column without $NaN_3$ started to decrease from 1.0 at 50.6 spore volumes (10.5 d) in comparison with the stabilization at $\approx 1$ in the column with $NaN_3$. Different from iron fillings column without $NaN_3$, nitrate completely broke through at 27.0 pore volumes (5.6 d) in iron fillings column with $NaN_3$. The difference indicated that microbial denitrification dominated the nitrate removal after certain amounts of nitrate input [25]. The average nitrate removal capacity within 750 pore volumes was 2670 mg-N/kg for limestone, which was 1.9 times higher than that for iron fillings (1400 mg-N/kg). This difference is attributed to the lower pH environment in limestone (7.65), which is more suitable for mircorganism growth than in the iron filings of higher pH (8.5–10.7) [33,38]. Microbial denitrification of nitrate enhanced nitrate removal capacity and extended the longevity of the reactive materials.

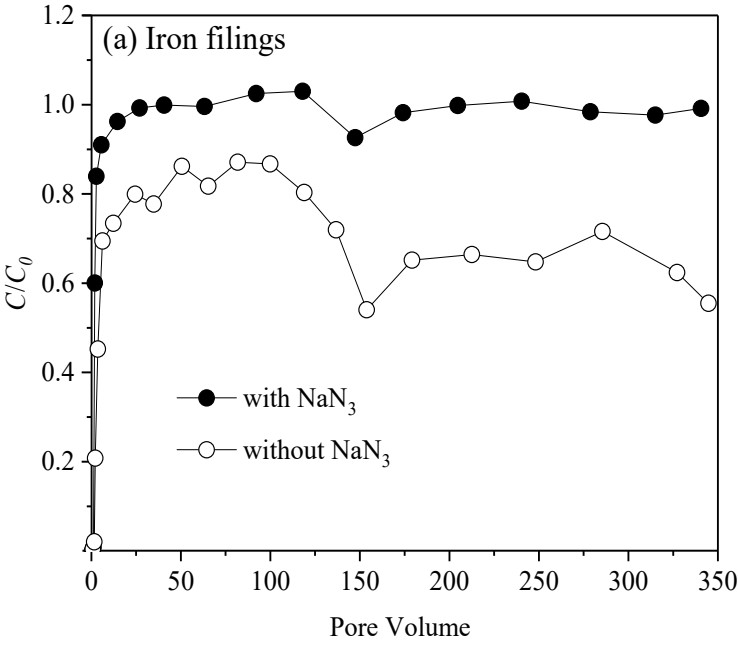

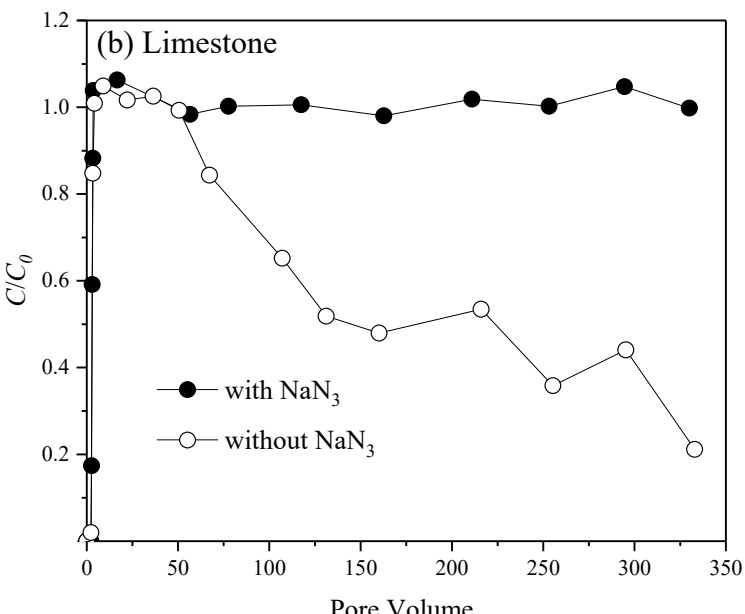

**Figure 6.** Effect of NaN$_3$ on nitrate breakthrough from the columns packed with (**a**) iron fillings and (**b**) limestone.

### 3.2. Removal of Phosphate

The phosphate breakthrough results of columns packed with quartz sand, iron filings and limestone are presented in Figure 7. The phosphate breakthrough from the quartz sand column occurred after the first pore volume, with $C/C_0$ increasing quickly to ≈1 at ≈2 pore volums and remaining stable in the following 173 pore volumes (36.5 d). This result shows no removal of phosphate by quartz sand.

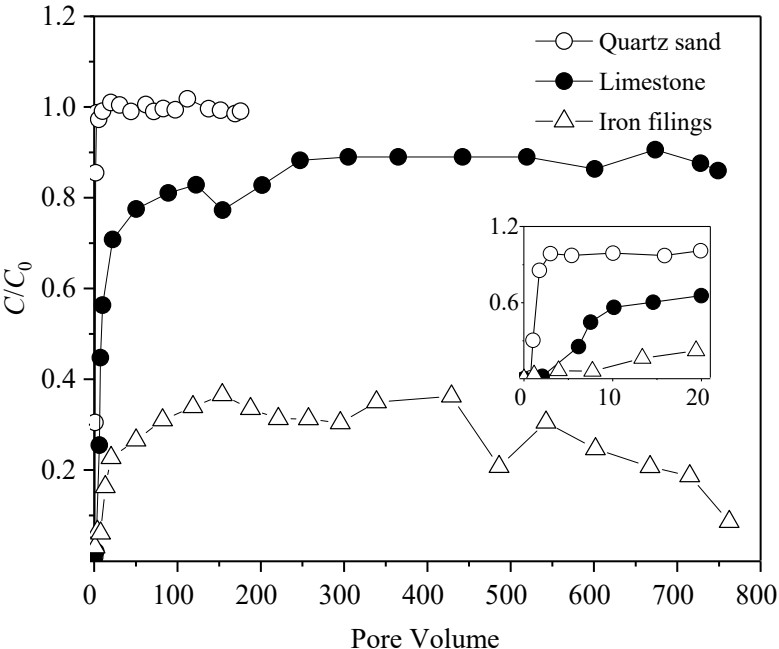

**Figure 7.** Phosphate breakthrough from the columns packed with different reactive materials. Subgraph is detailed view of the results within the first 20 pore volumes.

The phosphate $C/C_0$ of the limestone packed column increased with pore volume until stabilization at 0.88 at 247 pore volumes (51.5 d). The cumulative removal rate of phosphate by limestone was 17.6%, yielding an average phosphate removal capacity of 155.2 mg/kg within 750 pore volumes (156.3 d) (Table 2). Previous research has shown that natural mineral calcite ($CaCO_3$ with purity of 98.2%) can effectively remove 80–100% of phosphate at basic pH in batch experiments [39]. Electrostatic and chemical interactions were mainly responsible for phosphate removal [39]. It was reported that the effect of pH on phosphate removal by calcite can be attributed to the change in speciation of phosphorus and/or the surface sites of calcite [40]. In our study, the pH of effluent was 7.65 ± 0.41 compared to the influent pH of 6.7 ± 0.2. Due to $CaCO_3$ hydrolysis reaction at pH smaller than 8.2 (i.e., zero point of charge of calcite), cationic species (e.g., $Ca^{2+}$, $CaHCO_3^+$, and $CaOH^+$) were prevailing on the limestone surfaces, rendering the mineral surface positively charged [39]. Meanwhile, $H_2PO_4^-$ and $HPO_4^{2-}$ were dominant species at pH of 7.65 ± 0.41. As a result, the positively charged surface of limestone adsorbed $H_2PO_4^-$ and $HPO_4^{2-}$ via electrostatic attraction. In addition, phosphate could be removed through formation of complex compounds on the limestone during the following reactions occurring on the limestone surface [39–41].

$$Ca^{2+} + H_2PO_4^- \rightarrow CaH_2PO_4^+ \tag{8}$$

$$Ca^{2+} + HPO_4^{2-} \rightarrow CaHPO_4 \tag{9}$$

$$2Ca^{2+} + HPO_4^{2-} + HCO_3^- \rightarrow Ca_2HPO_4CO_3 + H^+ \tag{10}$$

$$Ca_2HPO_4CO_3 \rightarrow Ca_2PO_4CO_3 + H^+ \tag{11}$$

Similar to limestone, the phosphate $C/C_0$ of iron filings packed columns increased with time (Figure 7). The $C/C_0$ values became stable between 119 pore volumes (24.8 d) and 450 pore volumes (98.8 d) and even lower afterwards, demonstrating that iron filings could steadily remove phosphate for a relatively long duration. The effluent phosphate $C/C_0$ from iron filings plateauted at 0.34 at 119 pore volumes (24.8 d).This earlier stabilization and lower $C/C_0$ compared to limestone (0.83 at 24.7 pore volumes) suggests that iron filings had a longer life time and higher capacity for phosphate removal than limestone. The cumulative phosphate removal rate of iron filings was 68.2%, which was

≈4 times higher than that of limestone (17.6%). However, Sleiman et al. [26] reported that phosphate $C/C_0$ stabilized at 0.65–1.0 in iron/sand packed columns with different preconditioning durations, resulting in lower phosphate removal efficiency (20–34%). Such lower removal rates were likely due to the shorter hydraulic residence time and passivation of iron surface via pre-oxidation compared to this study. The phosphate average removal capacity of iron filings was 416.1 mg/kg, which was ≈3 times higher than that of limestone (155.2 mg/kg) (Table 2). Similar results were reported by Sellner et al. [14] that five industrial by-products (1.68–4.95 mg/g), including iron filings, steel slag, and three steel chips, exhibited approximately one order of magnitude higher phosphate adsorption capacity than that of three nature minerals (0.10–0.13 mg/g) including limestone, calcite, and zeolite at 20 °C in batch experiments. The phosphate removal by iron filings was attributed to Fe–P precipitation and P sorption on iron filings and its corrosion products [22,42]. Additionally, iron corrosion increased the specific surface area of iron filings, providing more adsorption sites for phosphate. The corrosion of iron led to an initial higher effluent pH of 10.7 [43]. This suggests that the iron corrosion could cause release of iron ions into the solution to form Fe–P precipitates as described below [13,22].

$$3\ Fe^{2+} + 2\ PO_4^{3-} \rightarrow Fe_3(PO_4)_2\downarrow \tag{12}$$

$$Fe^{2+} + n\ H_nPO_4^{(3-n)-} \rightarrow Fe(H_nPO_4)_n\ (n = 1,2) \tag{13}$$

$$Fe^{3+} + 2\ PO_4^{3-} \rightarrow FePO_4\downarrow \tag{14}$$

$$Fe^{3+} + H_nPO_4^{(3-n)-} \rightarrow [FeH_nPO_4]^{n+}\ (n = 1,2) \tag{15}$$

Previous research showed that iron corrosion could generate lepidocrocite, goethite, magnetite, and green rust I/II [26,44]. The corrosion might be responsible for gradual decrease in the effluent pH to 8.5 after 132 pore volumes (27.5 d) of the continuous input of experimental solutions. Similar to the surface of iron filings, these corrosive products adsorbed phosphate from the solution via electrostatic attraction and ligand binding [26,44]. These results are supported by the SEM (Figure 8) and EDX observations (Table 3), which demonstrate large variations of surface morphologies on the iron filings samples after phosphate adsorption compared to the smooth and uniform surfaces of the original iron filings (Figure 8). The post-iron filings surfaces were almost completely covered by a pervasive layer mainly made up of Fe and O with a small amount of P (Table 3). Compared with the original iron filings material, the content of Fe decreased while the contents of P and O increased after the transport experiments, confirming the formation of iron oxides, Fe–P precipitation, and phosphate sorption on the surfaces of iron hydroxides or iron oxides. These findings are in accordance with the change in the surface morphology of reacted iron in the Fe/$H_2O$ system for phosphate removal [26].

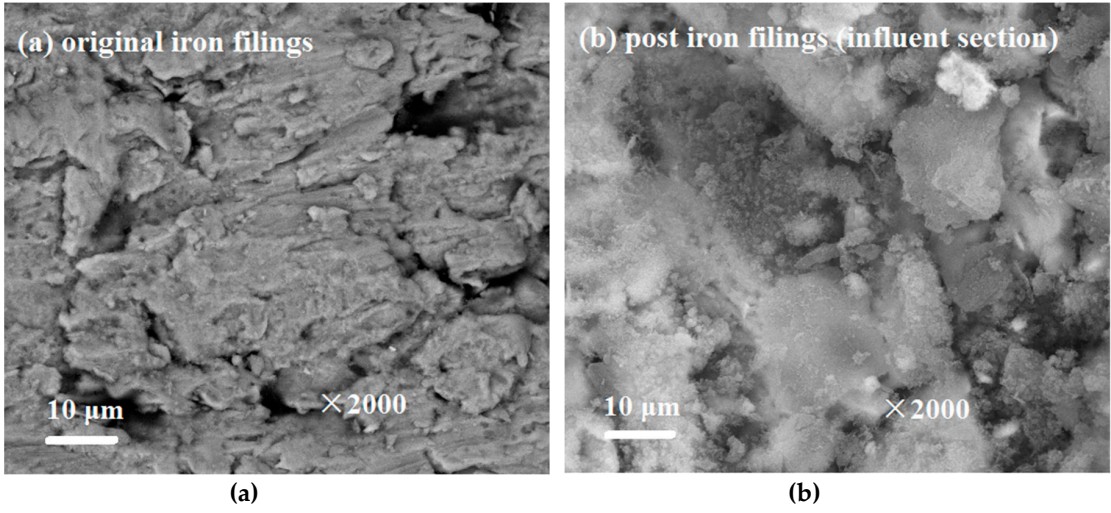

**(a)**                    **(b)**

**Figure 8.** *Cont.*

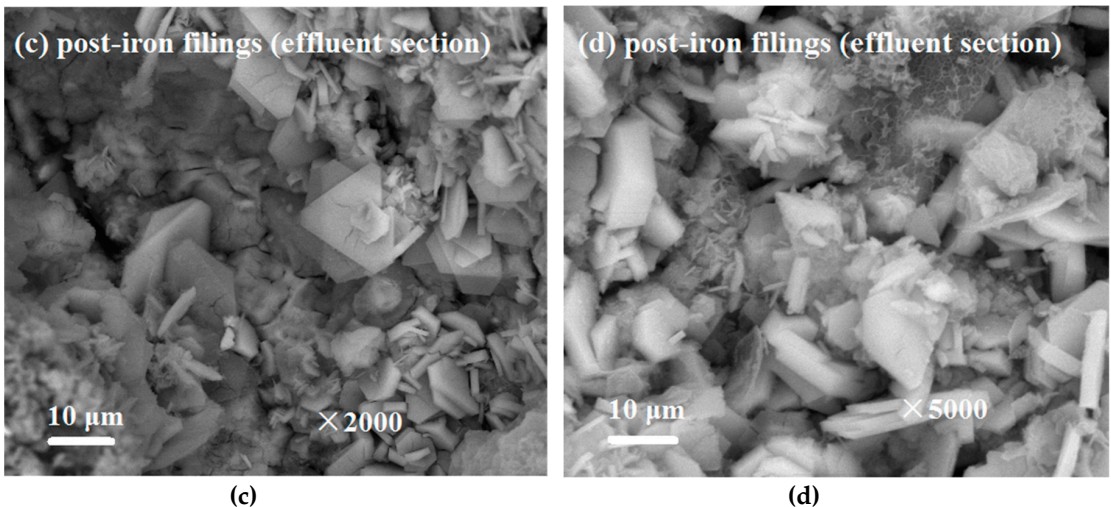

**Figure 8.** SEM images of (**a**) the original iron fillings, (**b**) post- experiment iron fillings collected from the influent section of column, and (**c,d**) post- experiment iron fillings collected from the effluent section of column.

**Table 3.** EDX analysis of elements on various iron fillings samples.

| Sample | Elements (wt%) | | | | | | | | |
|---|---|---|---|---|---|---|---|---|---|
| | **Fe** | **O** | **P** | **N** | **C** | **Ca** | **Si** | **Al** | **Mg** |
| Original iron fillings | 80.89 | 3.54 | 0.65 | 0.93 | 9.76 | 0.16 | 2.88 | 0.70 | 0.49 |
| Post- iron fillings (influent) | 75.45 | 11.33 | 1.30 | 1.01 | 7.78 | 0.17 | 2.00 | 0.68 | 0.27 |
| Post- iron fillings (effluent) | 65.59 | 19.47 | 0.87 | 1.17 | 8.80 | 0.14 | 2.44 | 0.91 | 0.62 |

Phosphate removal capacities in limestone (155.2 mg/kg) and iron filings (416.1 mg/kg) were underestimated because phosphate $C/C_0$ did not reach 1.0 during the column experiments, as shown in Figure 7. With influent continually input, $CaCO_3$ hydrolysis reaction and iron corrosion (various forms of iron oxides) can generate new surface sites for phosphate removal [41,44]. This self-regeneration capability of limestone and iron filings is beneficial for long-term phosphate capture in agricultural drainage. The additional phosphate adsorption sites will be continuously generated as the hydrolysis and corrosion reactions occurred on the surface of limestone and iron filings, thereby maintaining phosphate removal capacity and extending the longevity of reactive materials.

### 3.3. Removal of Pesticides

The pesticides breakthrough results of columns packed with quartz sand, iron fillings and limestone are demomstrated in Figure 9. The three pesticides had similar transport behaviors through quartz sand, with breakthrough occurring at ≈1 pore volume. The cumulative removal rates were 0%, 11.1%, and 17.4% for tricyclazole, isoprothiolane, and malathion, respectively, at 175 pore volumes (36.5 d). The average removal capacity were 0 mg/kg for tricyclazole, 4.7 mg/kg for isoprothiolane, and 5.2 mg/kg for malathion within 175 pore volumes (36.5 d) (Table 2). This result showed no removal of tricyclazole and small removal of isoprothiolane and malathion by quartz sand during saturated flow.

The pesticide breakthrough curves from iron filings and limestone shifted to the right, suggesting significant removal of the pesticides. Specifically, tricyclazole broke through limestone and iron filings columns at ≈1 pore volume and 8.1 pore volumes (1.7 d), respectively, and then their $C/C_0$ values increased to ≈1 and stabilized, with a cumulative removal rate of 1.1% in limestone and 22.0% in iron filings. The average removal capacity of tricyclazole was 30.1 mg/kg in iron filings, which was 15 times higher than that of limestone (2.2 mg/kg) (Table 2). The higher removal rate in iron filings is due that Fe iron enhanced the π–π interactions with the aromatic electron π cloud and the

electron lone pairs of S atom of tricyclazole [12]. Moreover, a larger specific surface area of post-iron filings than post-limestone provided more active sites for stronger interaction between tricyclazole and iron filings. Azarkan et al. [12] found that clay from the Dchiriyine zone in Tetouan (CT) had an adsorption capacity for tricyclazole (≈30 mg/kg) similar to the iron filings, but much lower than the adsorption capacity (620 mg/kg) of Ghassoul clay (GC). This difference is attributed to the higher specific surface area of GC (119 m$^2$/g) than CT (31 m$^2$/g) [12], iron filings (0.25–36 m$^2$/g), and limestone (0.14–0.28 m$^2$/g). Isoprothiolane had similar cumulative removal rate (27.0%) in limestone and iron filings, with the breakthrough occurring at ≈60 pore volumes. The average removal capacity of isoprothiolane was 51.8 mg/kg in iron fillings and 35.2 mg/kg in limestone (Table 2). Malathion breakthrough in limestone occurred at 280 pore volumes (58.3 d), whereas no breakthrough was detected in iron filings. The removal rates of malathion were 100% in iron filings and 91.6% in limestone (Table 2). Malathion removal capacities in iron filing (138.9 mg/kg) and limestone (166.8 mg/kg) were underestimated because the $C/C_0$ values of malathion did not reach 1.0 during the transport experiments as shown in Figure 9. Aydin [45] reported that the maximum malathion adsorption capacity of Fe$_3$O$_4$/red mud nanoparticles was 3333 mg/kg as calculated from batch experiments data.

The highest removal rates of malathion in limestone (91.6%) and iron filings (100%) attributed to the possession of hydrophilic double bonds and oxygen atoms of malation, which facilitated H-bonding and electrostatic interactions with the dangling OH groups of the external surface of materials [21]. Additionally, malathion possesses a nucleophilic sulfur functional group, which can covalently bind on carbonyl and other functional groups on the surfaces of iron filings and limestone [46]. Isoprothiolane was removed partly (27.0%) by both iron filings and limestone, whereas 22.0% and 1.1% of tricyclazole by iron filings and limestone, respectively. Similar to malathion, isoprothiolane has hydrophilic double bonds and oxygen atoms, which can bind on function groups containing hydrogen of materials through hydrogen bond [47]. However, tricyclazole is completely aromatic with a high electron delocalization through the benzene ring and has no polar functional groups [12], leading to unfavorable interactions between tricyclazole and porous materials. Overall, tricyclazole has greater risk for water bodies than isoprothiolane and malathion, due to its high mobility in the reactive materials.

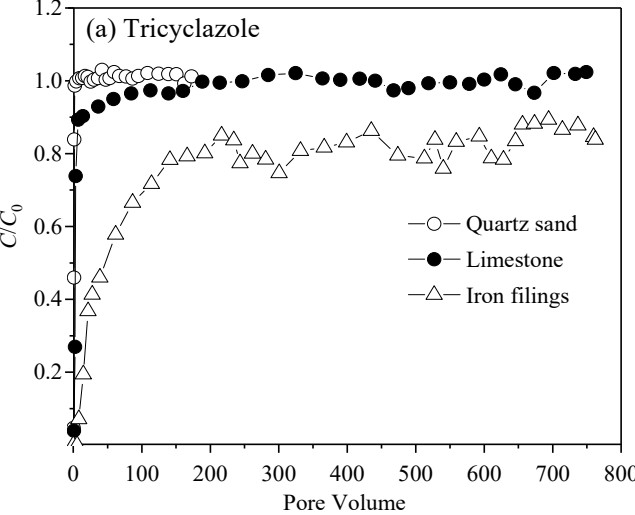

**Figure 9.** *Cont.*

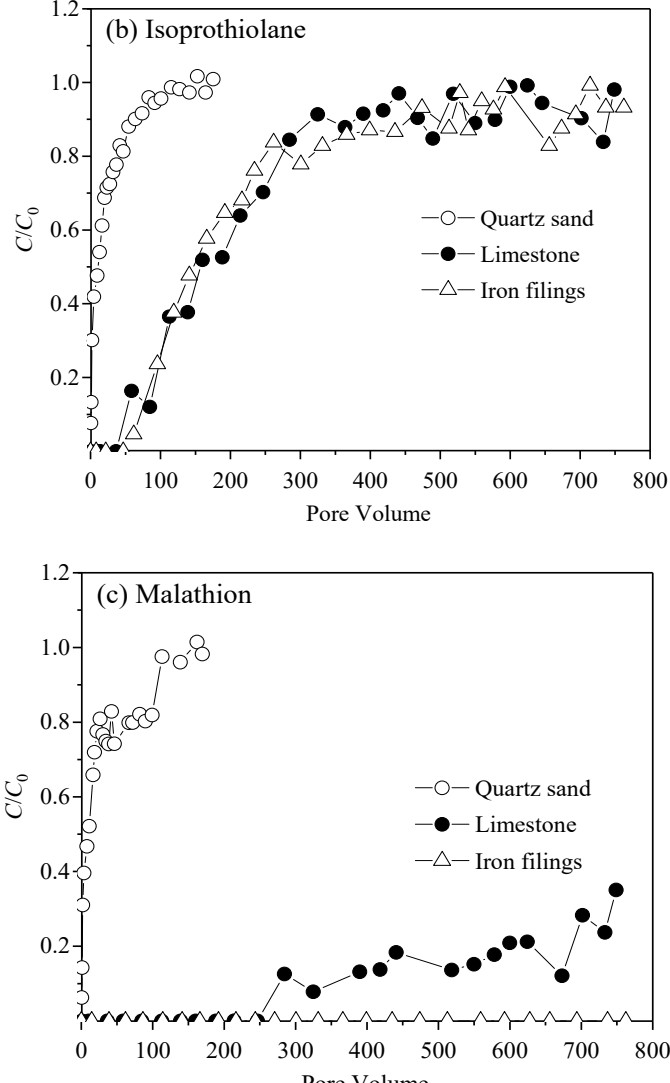

**Figure 9.** Pesticide breakthrough from columns packed with different reactive materials, (**a**) Tricylazole, (**b**) isoprothiolane, and (**c**) malathion.

## 4. Conclusions

One natural mineral and one industrial by-product were evaluated for simultaneous removal of phosphate, nitrate, and pesticides under flow conditions. Limestone showed higher nitrate removal rate (45.0%) than iron filings (35.8%) within 156.3 d, with average removal capacity of 2670 mg/kg in limestone and 1400 mg/kg in iron filings. While reduction to ammonia dominated nitrate removal in iron filings (19.3%), electrostatic attraction contributed to minor nitrate removal (3.7%) in limestone during early phase (i.e., <21.7 d in iron filing and <10.5 d in limestone). Iron filings and limestone may require relatively long operating times (e.g., >32.0 and 27.3 d, respectively) to fully develop the denitrification microbial community for nitrate removal. Breakthrough results showed that the phosphate removal rate (68.2%) of iron filings was 3.8 times higher than that of limestone (17.6%) within 156.3 d. Changes in morphology and crystallographic structures of experimental materials suggested that sorption, ligand exchange, and chemical precipitation were responsible for phosphate removal. Iron filing also exhibited higher removal rate for tricyclazole (22.0%) and malathion (100%) than limestone for tricyclazole (1.1%) and malathion (91.6%). This study provides important insights into the long-term potential for industrial wastes and natural minerals for concurrent removal nutrients

and pesticides from agricultural drainage under flow conditions. Future research should investigate the life time of the reactive materials and their combined efficiency under unsaturated flow condition.

**Author Contributions:** Conceptualization, X.C.; Methodology, X.C., J.Z., and D.T.; Data Gathering D.T.; Data Analysis D.T.; Validation, X.C.; Writing—Original Draft Preparation, D.T.; Writing—Review and Editing, X.C. and J.Z.; Project Administration, X.C. and J.Z.; Funding Acquisition, X.C. and J.Z.

**Funding:** This study received financial support from the National Natural Science Foundation of China (41671229, 41730858) and the National Key Research and Development Program of China (No. 2018YFC1801200).

**Conflicts of Interest:** The authors declare no conflicts of interest.

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
