# Peer review of "Reactive Transport and Removal of Nutrients and Pesticides in Engineered Porous Media"

_water, doi:10.3390/w11071316_

Round 1
Reviewer 1 Report
In this manuscript, the authors investigated the reactive transport and removal of nutrients and pesticides through porous media. They set up flow-through column experiments to explore the potential and mechanisms of selected materials to remove nitrate, phosphate and pesticides from synthetic agricultural drainage. The experiments are well designed and the results are clearly presented.
I have no real criticisms of the work itself, but do think the authors can move the figures & tables from the supplement material to the main text to help the readers easily understand the setup of the experiments. Note that the primary part of this work is experiments and there is no length limit of the manuscript published in the journal of "Water".
Author Response
Language expressions have been edited thoroughly throughout the manuscript. Following the suggestion, we have also moved the figures and tables from the supplement material to the main text and all figures have been numbered consecutively in accordance with their appearance in the text.
Reviewer 2 Report
The paper is very well written and includes original results
It can be published as it is
Author Response
Thanks for the nice comment.
Reviewer 3 Report
Reactive transport and removal of nutrients and pesticides in engineered porous media
In this study, the reactive transport and removal of nitrate, phosphate and selected pesticides by two low cost reactive materials, industrial wastes iron filings and natural ore limestone, is investigated by using column transport experiments. This is an experimental work.
Some useful results are presented. This paper is well prepared and can be recommended after some revisions.
1. How about the effect of pore structure of porous media on reactive transport and removal of nutrients and pesticides?
2. Could you add some mathermatical analysis or modeling for your data?
3. Could you compare some available works to your current paper?
Author Response
Our specific response to each comment is provided in the attached file.

Reviewer 4 Report
The authors analyse the removal process of three pesticides from two low-cost reactive materials. The authors investigate experimentally the removal capacity with the aid of a column experiment.
The problem is well introduced also for the non-expert readers. Methodology and results are carefully described. The english seems appropriate, but I am not a native english speaker.
The manuscript seems appropriate for the journal and can be accepted in the present form.
Author Response
Thanks for the nice comment. Language expressions have been edited thoroughly throughout the manuscript.